# Robust and bias-free localization of individual fixed dipole emitters achieving the Cramér Rao bound for applications in cryo-single molecule localization microscopy

Fabian Hinterer[1]☯, Magdalena C. Schneider[2]☯, Simon Hubmer[3], Montserrat López-Martinez[2], Philipp Zelger[4], Alexander Jesacher[4], Ronny Ramlau[1,3]*, Gerhard J. Schütz[2]*

**1** Institute of Industrial Mathematics, Johannes Kepler University Linz, Linz, Austria, **2** Institute of Applied Physics, TU Wien, Vienna, Austria, **3** Johann Radon Institute Linz, Linz, Austria, **4** Division for Biomedical Physics, Medical University of Innsbruck, Innsbruck, Austria

☯ These authors contributed equally to this work.
* ronny.ramlau@jku.at (RR); schuetz@iap.tuwien.ac.at (GJS)

**Data Availability Statement:** All relevant data are within the paper and its Supporting information

## Abstract

Single molecule localization microscopy (SMLM) has the potential to resolve structural details of biological samples at the nanometer length scale. Compared to room temperature experiments, SMLM performed under cryogenic temperature achieves higher photon yields and, hence, higher localization precision. However, to fully exploit the resolution it is crucial to account for the anisotropic emission characteristics of fluorescence dipole emitters with fixed orientation. In case of slight residual defocus, localization estimates may well be biased by tens of nanometers. We show here that astigmatic imaging in combination with information about the dipole orientation allows to extract the position of the dipole emitters without localization bias and down to a precision of 1 nm, thereby reaching the corresponding Cramér Rao bound. The approach is showcased with simulated data for various dipole orientations, and parameter settings realistic for real life experiments.

## Introduction

The rapid progress of superresolution microscopy in the last 15 years has enabled fluorescence microscopy well below the diffraction limit of light. Under appropriate imaging conditions, a resolution of around 20 nm can be routinely achieved in biological samples [1]. One prominent group of methods, termed single molecule localization microscopy (SMLM), exploits the stochastic blinking of single dye molecules to circumvent the diffraction barrier [2]. By determining the positions of all dye molecules to a precision far below the wavelength of light, localization maps can be obtained, which represent a superresolved image of the biological sample. Since the image acquisition requires the recording of ten thousands of frames, easily lasting tens of minutes, appropriate sample fixation has become inevitable. While most chemical

files. In addition, we included the code for simulation of PSF images and the fitting procedure. Thus, all data can be recreated via the software. We provided the link to the software package in the Supporting information Section.

**Funding:** The authors were funded by the Austrian Science Fund (FWF): F6805-N36 (RR), F6809-N36 (GJS), P30214-N36 (AJ). The funders had no role in study design, data collection and analysis, decision to publish, or preparation of the manuscript.

**Competing interests:** The authors have declared that no competing interests exist.

fixation protocols provide sufficient quality for the needs of diffraction-limited microscopy, SMLM demands for improved preservation of biological structures [3].

Cryo-fixation currently represents the gold standard for this task [4], and in particular outperforms chemical fixation procedures which are frequently affected by residual protein mobility [5] or morphological changes of the sample [6]. Importantly, cryo-fixation is compatible with SMLM [7–10]. As an additional advantage, it has been established that vitrification at cryogenic temperatures improves the photostability of fluorophores and gives access to much higher resolution through an increase in the signal to noise ratio [7]. However, compared to samples imaged at ambient temperatures, the localization procedure becomes more complicated: at room temperature dipoles are freely rotatable, yielding on average a Gaussian-like point-spread function (PSF), whereas at low temperatures one has to account for the PSF pattern caused by fixed dipoles. In addition, light microscopy at cryogenic temperatures typically prohibits the use of high numerical aperture (NA) oil-immersion objectives. Instead, one is restricted to using a long-distance air-objective, resulting in much lower NA, typically 0.7 − 0.8.

In SMLM, two-dimensional fluorophore positions are usually determined by fitting an isotropic two-dimensional Gaussian function. Using this Gaussian approximation achieves excellent results in the case of freely rotating fluorophore dipoles and is computationally very efficient [11, 12]. However, for fixed dipoles the PSF becomes distorted and tilted with respect to the optical axis (Fig 1b), yielding biased estimations of the actual fluorophores' positions [13]. In particular, it has been noted that for molecules located away from the focal plane, the lateral shift of the recorded PSFs can easily reach 100 nm for tilted dipoles [14]. The size and direction of this shift depends on the amount of defocus as well as on the molecular orientation (Fig 2). It is therefore not a unique shift that could easily be corrected. In summary, using a Gaussian approximation of the PSF will result in substantial localization bias in the case of fixed dipoles [11, 12], and image formation models more sophisticated than a simple Gaussian are required if one wants to avoid these errors. Indeed, PSF models based on the Kirchhoff vector approximation [15] or Hermite functions [16] for a dipole point source yielded unbiased estimations of the localization. However, the optical systems consisted of a high-NA objective, which does not comply with the cryogenic approach discussed here. In contrast, for low NA objectives the PSF of a defocused fixed dipole is in general hardly distinguishable from one that is laterally shifted. Particularly for high levels of background noise, it turns out to be virtually impossible to estimate the correct position from a single image, hence leading to instable fit results.

In this manuscript, we show that by utilizing additional information on the defocus and on the dipole orientation one can improve the stability of the fit, yielding unbiased and precise localization results. A well-established method which allows for defocus estimation is to introduce a weak cylindrical lens in the optical path, resulting in an elliptical distortion of the PSF of an emitter above or below the focal plane [17] (compare Fig 1b). Astigmatism-based super-resolution microscopy is widely used for estimating the z-position of isotropic PSFs [18–20]. In order to determine the dipole orientation, two categories of approaches can be distinguished: (i) In spot shape analysis, the delicate way in which the dipole orientation of a fluorophore affects the shape of the PSF (e.g. [21]) is utilized to derive its orientation. Many methods rely on a comparison of the measured PSF with pre-computed templates. We refer the reader to [13] for an extensive overview of methods of this class. However, as mentioned above such methods require the use of a high-NA objective. In the case of low-NA microscopy, the PSFs are radially symmetric, making a reconstruction of the azimuthal angle purely based on spot-shape infeasible. (ii) One may exploit the polarization effects in absorption or emission. Excitation with linearly polarized light results in excitation yields proportional to $\cos^2(\beta)$, where $\beta$ is

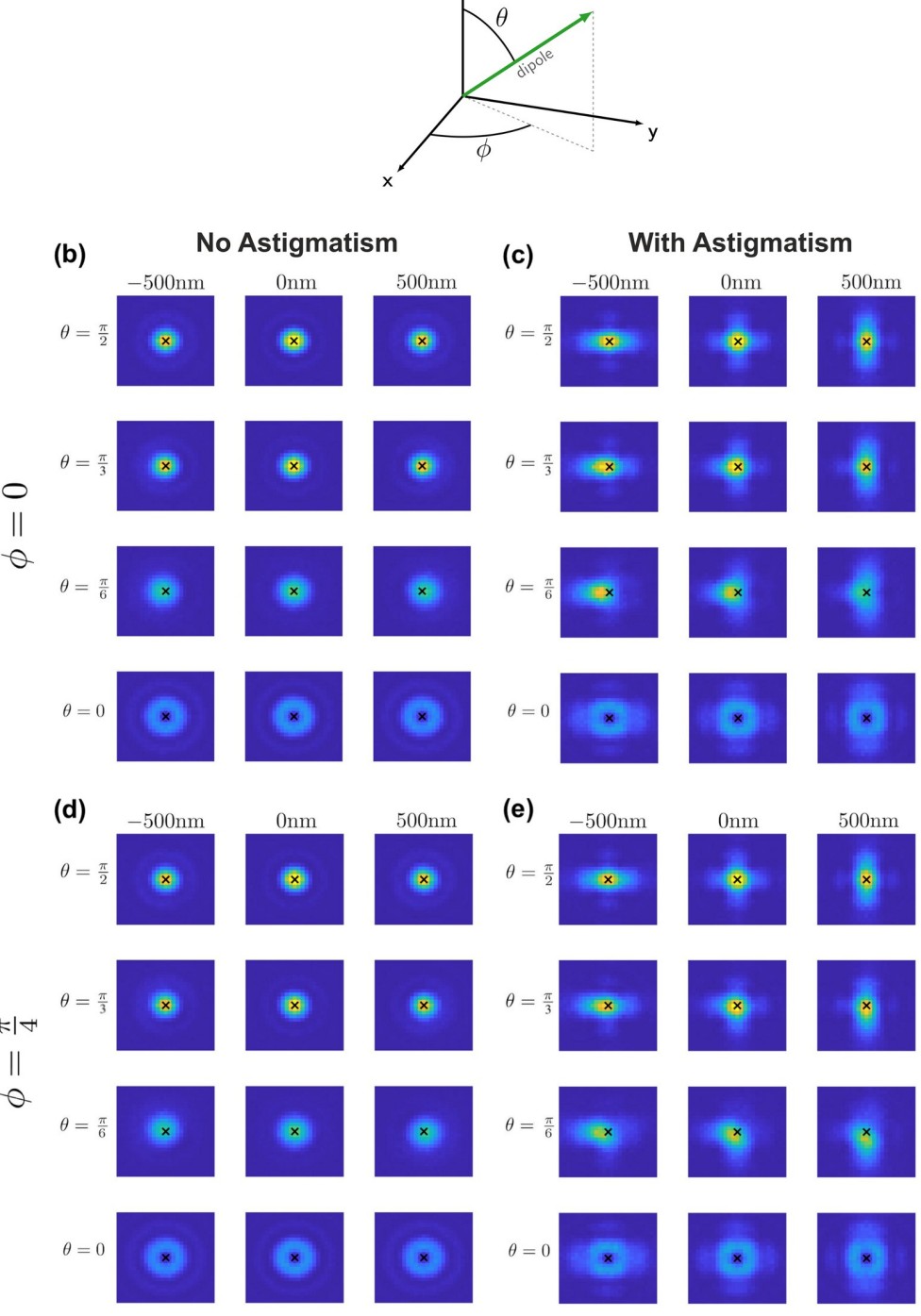

**Fig 1. PSFs for different dipole orientations.** (a) Illustrative dipole orientation with azimuthal angle $\phi$ and inclination angle $\theta$. The optical axis is assumed to be parallel to the $z$-axis. PSF intensity patterns with no astigmatism (b,d) and 75 nm RMS vertical astigmatism (c,e). The inclination angle is set to $\theta = \pi/2$, $\pi/3$, $\pi/6$, 0 and the azimuthal angle is $\phi = 0$ (b,c) and $\phi = \pi/4$ (d,e). The black cross indicates the actual position of the dipole emitter.

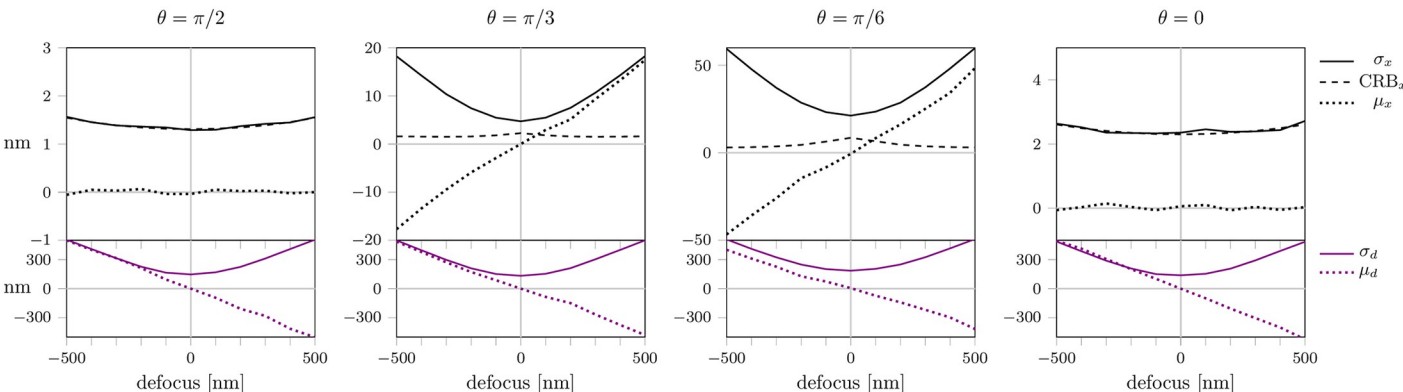

**Fig 2. Localization errors without astigmatism.** We fitted the position and defocus $(x, y, d)$, while the dipole orientation $(\theta, \phi)$ was assumed to be known exactly. Shown are localization precision $\sigma_x$, $\sigma_d$ (solid lines) and bias $\mu_x$, $\mu_d$ (dotted lines) for the $x$-position (black) and defocus values $d$ (violet), respectively, that arise in astigmatism-free imaging for dipole orientations with inclinations angle $\theta = \pi/2$, $\pi/3$, $\pi/6$, 0 and azimuthal angle $\phi = \pi/4$. For symmetry reasons, localization precision and bias for the $x$- and $y$-position are identical in the astigmatism-free case. The CRB for localization along the $x$-direction is indicated by the dashed line. The number of photons was set to $N = 5 \cdot 10^5$ and the background noise to $b = 300$. Each data point represents 1000 simulations.

the angle between the electric field and the absorption dipole moment. The amount of photons collected from a particular fluorophore at a specific polarization angle then allows to draw inferences about its orientation [22, 23]. For example, one may consider an experiment, in which the fluorophores are alternately excited with light of two orthogonal linear polarization directions. Then, the azimuthal angle $\phi$ of the dipole axis is specified by the ratio of two intensity values, and its elevation angle $\theta$ can be derived from the sum of the two intensities related to the maximum single molecule intensity (Fig 1a). On the other hand, one can split the emitted photons into different polarization channels [24–26]. Since many approaches for determining the dipole orientation are available which are applicable to the low-NA setting considered in this work, we will assume that an estimate of the dipole orientation is available.

In this paper, we introduce a maximum likelihood based approach to extract the two-dimensional position of non-rotating single molecule emitters, which is directly applicable to recordings at cryogenic temperatures. We show that the algorithm is bias-free and achieves a precision close to the Cramér Rao bound (CRB).

## Methods

### PSF model

The PSF model used in this work is based on the model described in [27]. It comprises the electric field originating from a dipole emitter, the propagation through the layers of the optical system and the resulting intensity distribution in the image plane. We additionally introduced vertical astigmatism into the optical system, breaking the symmetry of the PSF above and below the focal plane and allowing for an estimate of the defocus.

We assumed a standard optical setup consisting of a sample layer with refractive index $n_1 = 1.33$ and an immersion layer with refractive index $n_2 = 1$, reflecting the situation of an air objective corrected for the presence of a cryostat window. We chose a coordinate system in the canonic way, i.e. such that the $z$-axis coincides with the optical axis and the $xy$-plane coincides with the focal plane. A fluorophore situated in the focal plane can be described by a dipole point source with lateral position $(x, y)$ and orientation $(\theta, \phi)$, where $\theta$ and $\phi$ denote the inclination and azimuthal angle, respectively (see Fig 1a).

The starting point for our model is the electric field vector $E_{BFP}$ (in Cartesian coordinates) in the back focal plane given by Eq. (18) in [21]. The electric field vector $E_f$ in the image plane is then given by the Fourier transform (Eq. (5–14) in [28])

$$E_f(x_f, y_f) = \frac{1}{i\lambda f} e^{i\frac{\pi}{\lambda f}(x_f^2 + y_f^2)} \iint E_{BFP}(x, y) e^{i\frac{2\pi}{\lambda}W(x,y)} e^{-i\frac{2\pi}{\lambda f}(x_f x + y_f y)} dx dy, \quad (1)$$

where $f$ denotes the focal length of the tube lens, $\lambda$ the emission wavelength, and integration happens over the circular pupil area. The aberration term $W(x, y)$ introduces wavefront aberrations (Eq. (6–33) in [28]), which can be expanded into a linear combination of orthonormal Zernike polynomials, i.e.,

$$W(x, y) = \sum w_i Z_i(x, y), \quad (2)$$

where $Z_i$ denotes the $i$-th Zernike polynomial (using Noll's indices) and $w_i$ is the corresponding Zernike coefficient. For the calculation of the Zernike polynomials we used [29]. The normalized intensity distribution is then given by

$$I(x_f, y_f) = |E_f(x_f, y_f)|^2 \cdot \left( \iint |E_f(x_f, y_f)|^2 dx_f dy_f \right)^{-1}. \quad (3)$$

## Simulations

Simulations were carried out in MATLAB using implementations of the above equations on a discrete grid. For a given dipole emitter with position $(x, y)$ and orientation $(\theta, \phi)$ we calculated the intensity distribution (Eq (3)) within a region of interest (ROI) of size $17 \times 17$ pixels. Calculating the intensity values only at the discrete positions of the camera pixels may lead to inaccuracies. We therefore computed the values on a finer grid. The values of the smaller pixels were summed up to obtain the model value for each camera pixel. We chose an oversampling factor of 9. Further refinement of the discretization did not yield any measurable improvements (S1 Fig in S1 File).

For each simulation the position $(x, y)$ was chosen randomly in an area of $216 \times 216$ nm, typically corresponding to $2 \times 2$ pixels, in the center of the ROI.

In Eq (2), we only considered nonzero coefficients $w_4$ and $w_6$ corresponding to defocus and vertical astigmatism, respectively. For the coefficient $w_6$ we chose a value of $w_6 = 0.11\lambda$, corresponding to a RMS wavefront error of approximately 75 nm.

Unless specified otherwise, for our simulations we assumed the air objective ($n_2 = 1$) LUCPLFLN60X (Olympus), which has a magnification of 60x, a numerical aperture NA = 0.7 and a focal length of 3 mm. The focal length of the tube lens was set to $f = 180$ mm. For the sample we assumed dyes with an emission in the red region of the spectrum ($\lambda = 680$ nm), which were used to stain the biological sample with a refractive index of water ($n_1 = 1.33$). As detector we assumed a sCMOS camera with a pixel size of 6.5 μm, corresponding to 108 nm in object space.

For the dipole orientation, we considered combinations of the values $\phi \in \{\frac{\pi}{4}, 0\}$ and $\theta \in \{\frac{\pi}{2}, \frac{\pi}{3}, \frac{\pi}{6}, 0\}$. We simulated at defocus values ranging from −500 to 500 nm in steps of 100 nm. We applied Poissonian shot noise for each pixel and for some simulations additionally considered Poissonian background noise with standard deviation $b$.

The absorption probability depends on the fluorophore dipole orientation. Particularly, the detected numbers of emitted photons $N_x$, $N_y$ for excitation light with polarization direction

along the $x$- or $y$-axis, respectively, is given by

$$N_x = N_{\max} \sin^2(\theta) \cos^2(\phi) \,, \tag{4}$$

$$N_y = N_{\max} \sin^2(\theta) \sin^2(\phi) \,. \tag{5}$$

Here, $\theta$ and $\phi$ are the inclination and azimuth angle of the fluorophore's dipole relative to the $x$-axis, and $N_{\max}$ is the number of photons for parallel dipole orientation and polarization vector of the excitation light. The total number of detected photons $N_{\mathrm{eff}}$ is thus given as

$$N_{\mathrm{eff}} = N_x + N_y = N_{\max} \sin^2(\theta). \tag{6}$$

If not specified otherwise we assumed $N_{\max} = 5 \cdot 10^5$ photons.

## Cramér-Rao bound

The variance of any unbiased estimator $\hat{\xi}$ of a parameter vector $\xi$ is bounded from below by the Cramér-Rao Bound (CRB) [30]:

$$\mathrm{Var}(\hat{\xi}_k) \geq (\mathcal{I}^{-1}(\xi))_{kk}. \tag{7}$$

The CRB is given by the diagonal elements of the inverse Fisher information matrix for the underlying stochastic process which models the acquired data. The Fisher information matrix $\mathcal{I}(\xi)$ is given by

$$\mathcal{I}(\xi) := \mathbb{E}\left[ \left( \frac{\partial}{\partial \xi} \ln f_\xi(z) \right)^T \left( \frac{\partial}{\partial \xi} \ln f_\xi(z) \right) \right], \tag{8}$$

where the function $f_\xi$ denotes the probability distribution function of the data generation process. The vector $\xi$ consists of the parameters one wishes to estimate, which in our case are the lateral location and defocus, i.e. $\xi := (x, y, d)$.

An estimator is said to be efficient, if it attains the CRB. If an estimator is efficient, it fully utilizes the information which is contained in the data and the precision cannot be increased further. It has been shown that only a maximum likelihood estimator can attain the CRB [31].

In order to calculate the CRB, one first needs to choose an appropriate model which describes the image data. A Poissonian model has been demonstrated to be a reasonable approximation for photon shot noise [32]. The photon count $z_k$ in the $k$th pixel is modeled as the realization of a Poissonian random variable with mean $v_{\xi,k}$ and the probability distribution

$$f_{\xi,k}(z_k) = \frac{v_{\xi,k}^{z_k} e^{-v_{\xi,k}}}{z_k!} \,. \tag{9}$$

The probability distribution $f_\xi$ for the whole image is then given by the product

$$f_\xi(z) = \prod_{k=1}^{K} f_{\xi,k}(z_k) \,. \tag{10}$$

Combining Eqs (9) and (10) gives

$$\ln f_\xi(z) = \sum_{k=1}^{K} [z_k \ln(v_{\xi,k}) - v_{\xi,k} - \ln(z_k!)] \,. \tag{11}$$

The resulting Fisher information matrix is then given by

$$\mathcal{I}(\xi) = \sum_{k=1}^{K} \left(\frac{\partial v_{\xi,k}}{\partial \xi}\right)^{T} \left(\frac{\partial v_{\xi,k}}{\partial \xi}\right) \frac{1}{v_{\xi,k}} \, , \qquad (12)$$

where we refer to [32] for a detailed calculation. Finally, the CRB is obtained by taking the inverse of the matrix $\mathcal{I}(\xi)$ in Eq (12). Since analytical computation of the partial derivatives in (12) is infeasible, we do not compute the partial derivatives analytically, but approximate them by numerically computing difference quotients.

## Fitting

First, as an *a priori* estimator for the mean background signal $b^2$ we calculated the mean signal of an image without fluorophore signal. Next, we determined an estimate for the total number of detected photons $N_{eff}$ per molecule by summing over all noise-corrected pixels. For estimation of the parameters $\xi := (x, y, d)$, we used a maximum likelihood estimator, since only in this case the localization precision can attain the CRB [31]. Fitting was performed on normalized images. If not mentioned otherwise, the whole $17 \times 17$ region of interest was used for fitting. The log-likelihood function is identical to Eq (11); here $z_k$ denotes the photon number of the $k$th pixel of the normalized image. The fit function $v_{\xi,k}$ was calculated in the following way: we first determined the normalized PSF using Eq (3). The PSF was calculated on a discrete grid using an oversampling factor of 3, representing a good compromise between accuracy and computational speed (S1 Fig in S1 File). Further, the obtained PSF was multiplied by $N_{eff}$, and the mean of the background signal $b^2$ was added to each pixel. Finally, this function was normalized by the total sum of detected photons. The negative log-likelihood function was then minimized using the MATLAB function *fminunc*, yielding an estimate $\hat{\xi} := (\hat{x}, \hat{y}, \hat{d})$. It turned out to be important to select appropriate starting values, in order to avoid that the maximum likelihood estimator is trapped in local minima. Hence, we implemented a non-linear least squares fit, the outcome of which was used as the starting value for the maximum likelihood fit. The $(x, y)$ starting values for the non-linear least squares fit were chosen randomly within a $2 \times 2$ pixel region around the center of the image, and the starting value for the defocus in the interval $-500$ nm $< d < 500$ nm.

In case of astigmatic imaging, we assumed the astigmatism to be known for the fitting procedure. Also, for the fluorophore dipole orientation $(\theta, \phi)$, we assumed that an estimate $(\hat{\theta}, \hat{\phi})$ is available. We considered three different cases for the errors $\hat{\theta} - \theta$ and $\hat{\phi} - \phi$: (i) no errors, (ii) both errors in $\theta$ and $\phi$ are distributed normally with mean 0 and variance 2˚, (iii) the errors in $\theta$ and $\phi$ are distributed normally with mean 0, and variance 4˚ or 2˚, respectively.

For each parameter set, we simulated $n = 1000$ PSF images. We calculated the accuracy $\mu_x = \frac{1}{n}\sum_{i=1}^{n}(\hat{x}_i - x_0), n = 1000$, which corresponds to the bias of the localization procedure. We also calculated the precision $\sigma_x$ of the fitting procedure as the standard deviation of the error, i.e. $\sigma_x := \sqrt{\frac{1}{n-1}\sum_{i=1}^{n}\left((\hat{x}_i - x_0) - \mu_x\right)^2}, n = 1000$. The corresponding values $\sigma_y$ and $\mu_y$ were calculated analogously.

## Results

It was shown previously that the anisotropic emission of fixed dipole emitters can easily lead to substantial deviations between the fitted and the actual position of the molecule [33]. This is mainly due to the difficulty to extract the exact defocus value for each dipole emitter directly from the images. This issue becomes significant in case of noisy images, in which background

fluctuations mask faint differences in the shape of the PSF. Consequently, the fitting may be trapped in local minima, yielding a large influence of the starting values on the obtained fit results.

In Fig 2 we show a quantification of these errors upon fitting a point emitter with inclination angle $\theta = \pi/2, \pi/3, \pi/6, 0$. For these plots we assumed identical amounts of detected photons $N = 5 \cdot 10^5$ and background noise with a standard deviation $b = 300$. Indeed, for experiments performed under cryogenic temperatures the number of detected photons can easily exceed $10^6$ for individual fluorophores [9, 34]. Typical single frame noise levels are much smaller than $b = 300$. However, many researchers merge single frames before analysis, thereby increasing background noise. A noise level with a standard deviation $b = 300$ commonly allows the merging of more than hundred frames.

All simulated data were fitted with a maximum likelihood estimator using the theoretical model, yielding the parameters $\hat{\xi} = (\hat{x}, \hat{y}, \hat{d})$ (see sections *PSF model* & *Fitting*). To emulate a practical scenario, we assumed the molecules to be located within a range of ±500 nm around the focal plane, with the true defocusing value $d$ unknown. The choice of starting values for the fit reflects this scenario by selecting the starting value for $d$ randomly in the interval [−500 nm, 500 nm]. Theoretically, a maximum likelihood fit with the exact PSF model should yield bias-free results. However, the presence of background noise leads to highly unstable fit results, which depend strongly on the chosen starting values. In this particular case, bias values for the lateral localization $\mu_x$, $\mu_y$ up to 50 nm can be observed. Only in the symmetric scenarios $\theta = \pi/2$ and $\theta = 0$ the localization bias vanishes. For a reduced noise level ($b = 100$) the results are improved for small defocusing. However, in the case of large defocusing a localization bias of up to nm 50 remains (S2 Fig in S1 File). Note that the apparent absence of localization bias in case of $d = 0$ is a consequence of the symmetrically distributed starting positions assumed in our simulations, which compensate positive and negative bias values; these errors still affect the localization precision, which thus exceeds the CRB (dashed line).

In order to be able to determine the defocus, we considered an astigmatic imaging approach, which allows one to obtain $x$, $y$, and $d$ for the dipole emitter simultaneously. Such an approach is routine and available in many laboratories using single-molecule microscopy [18–20]. We assumed here a very weak astigmatism corresponding to a shift of the two foci of approximately 1.4 μm.

First, we fitted not only the lateral position ($x$, $y$) and the defocus value $d$, but also the dipole orientation ($\theta$, $\phi$), see Fig 3a. While this yielded unbiased fit results, the fit was unstable leading to high values of localization precision up to 40 nm. In order to increase the stability of the fit, we assumed the azimuthal angle $\phi$ to be known and left only the inclination angle $\theta$ as an open parameter for the fit (Fig 3b). The localization precision could indeed be improved, yet values of around 20 nm were still rather high. We conclude that the simultaneous reconstruction of position, defocus and orientation does not provide satisfactory results.

Subsequently, we fitted only the lateral position and defocus and assumed the dipole orientation to be known. First, we simulated fixed dipoles of different, precisely known orientations ($\theta$, $\phi$) for various defocus values and $N = 5 \cdot 10^5$ photons per PSF. In this situation, we ignored contributions of background noise ($b = 0$), i.e. photon shot noise presented the only source of noise. The resulting mean localization precision $\sigma$ and bias $\mu$ as well as the CRB are shown in Fig 4. As anticipated, the localization bias could be avoided. Also the localization errors were dramatically reduced, reaching the CRB over the whole range of defocusing. The precision in $x$ and $y$-direction showed opposing trends for a defocus in the range of −500 to 500 nm due to the different focal planes in $x$- and $y$-direction. Even though the PSF patterns and the positions of the intensity maxima depend on the azimuthal angle $\phi$ (see Fig 1), for both angles $\phi = \pi/4$

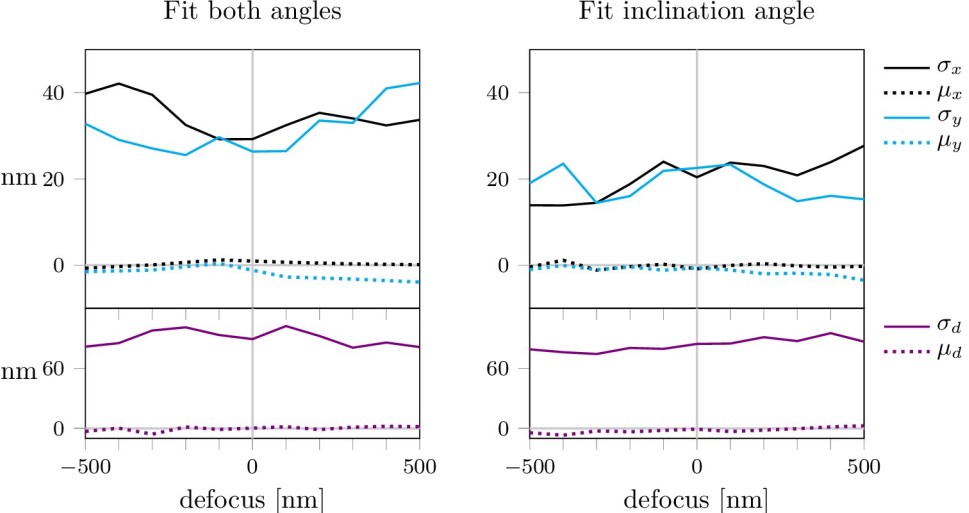

**Fig 3. Fitting dipole orientation with astigmatism.** We fitted the lateral position $(x, y)$, defocus $d$ and dipole orientation in the presence of astigmatism. In panel (a) we fitted both the inclination and azimuthal angle $(\theta, \phi)$. In panel (b) we assumed the azimuthal angle $\phi$ to be known and fitted the inclination angle $\theta$ only. Each data point represents 5000 simulations. For each simulation, the ground truth dipole orientation was chosen randomly. Background noise was set to $b = 100$.

and $\phi = 0$ the localization precision and bias were similar (S3 Fig in S1 File). Results for the same conditions, but reduced photon counts of $N = 5 \cdot 10^4$ or $N = 5 \cdot 10^3$ per PSF image, yield the same trends, but with increased localization errors (S4 Fig in S1 File). Also a twofold increase of the pixel size hardly affects the results (S5 Fig in S1 File).

In a real-life SMLM experiment, background signal arises from the presence of unspecific fluorescence as well as scattering. To account for this, we assumed homogeneous Poissonian-distributed background noise with a magnitude of $b = 100$. For all tested scenarios the obtained localization precision follows the CRB very well. Only in case of $\theta = \pi/6$ slight deviations can be observed, which can be attributed to errors in the estimation of the defocus. Essentially, a localization precision below 2 nm for $b = 100$ (Fig 5) and below 6 nm for $b = 300$ (S6 Fig in S1 File) can be expected.

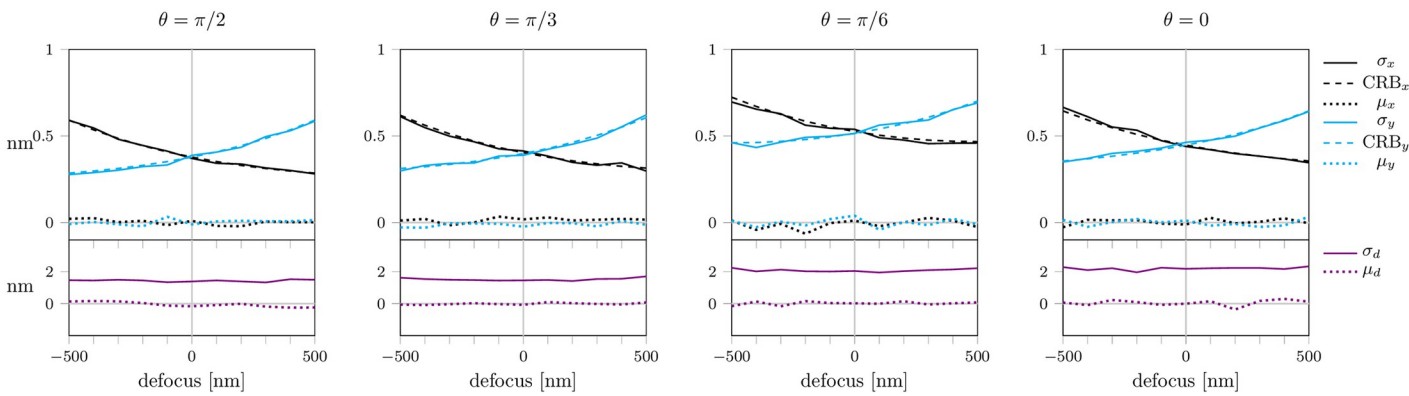

**Fig 4. Localization errors in the presence of astigmatism.** We fitted the position and defocus $(x, y, d)$, while the dipole orientation $(\theta, \phi)$ was assumed to be known exactly. Background noise was set to $b = 0$. All other parameters were identical to those of Fig 2. A list of all simulation parameters is given in S1 Table in S1 File.

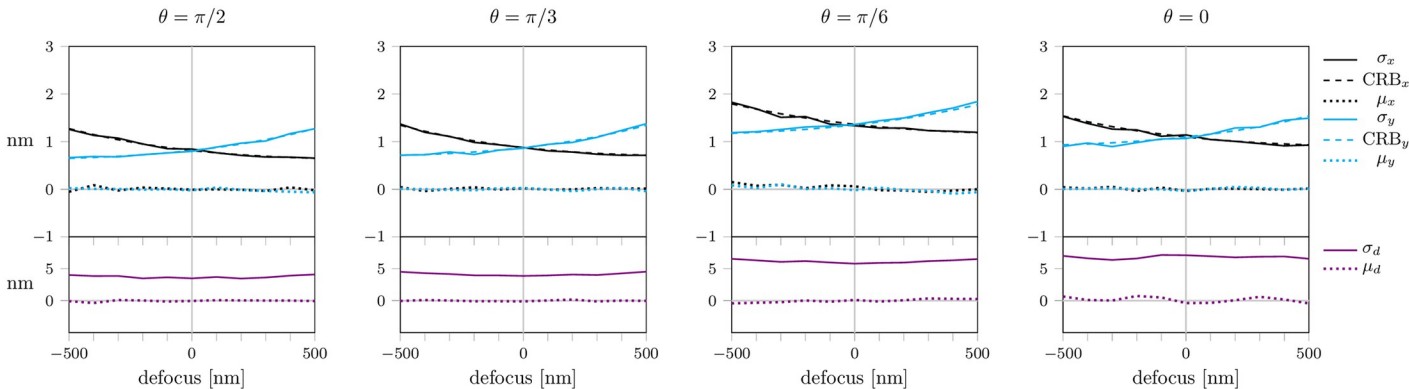

**Fig 5. Influence of background noise.** Simulations and fitting procedure were analogous to Fig 4, except for adding background noise with $b = 100$. A list of all simulation parameters is given in S1 Table in S1 File.

The obtained values for the precision and bias turned out to be surprisingly stable with respect to variations in the size of the analyzed ROI (Fig 6). We observed deviations from the CRB mainly for small sizes of the ROI, where the fit becomes more sensitive to slight changes in the subpixel position of the dipole emitter. This figure further confirms our choice of a 17 pixel ROI, which provides a good compromise between high precision and fitting stability.

In practice, exact knowledge about dipole orientation is unrealistic. Therefore, we investigated the effect of errors in the orientation estimation. We considered two different kinds of error distribution, which reflect realistic values [27]: (i) In Fig 7 and S7 Fig in S1 File, we considered the error to be normally distributed with mean 0 and standard deviation 2° for both angles $\theta$ and $\phi$. (ii) In S8 and S9 Figs in S1 File, we increased the error in the estimation of $\theta$ to a standard deviation of 4°, which reflects the higher difficulty of correctly estimating the inclination angle. For both cases of error distribution in dipole estimation, the fitting results yielded no bias. The trends for localization precision for the $x$- and $y$-axis were similar to the results for exact dipole estimation (compare Fig 4). However, the relative change in localization error between a defocus value of −500 and 500 nm increased. Substantial errors occurred

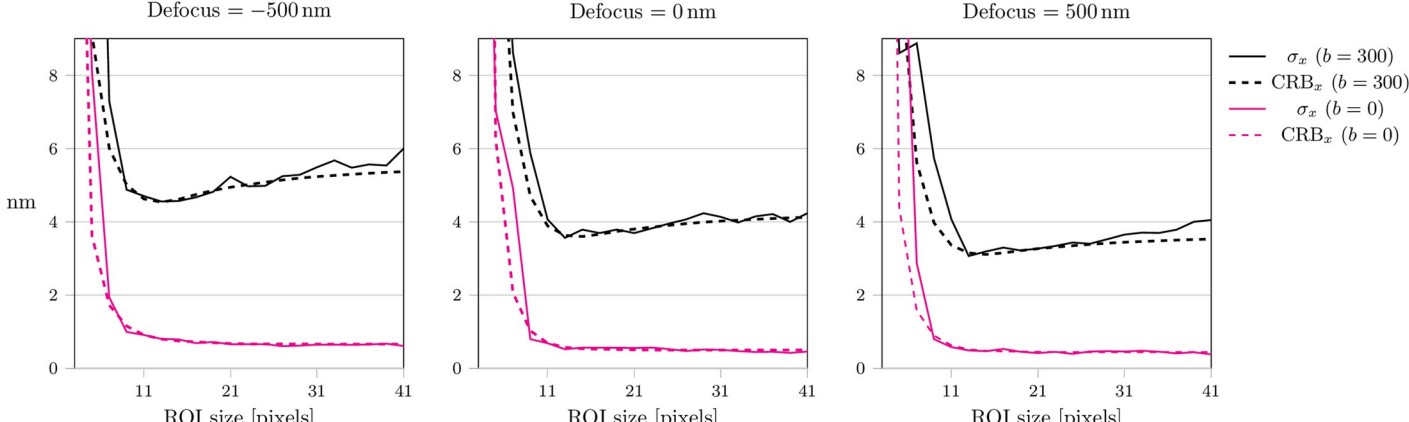

**Fig 6. Influence of the size of the fitted region.** Precision (solid lines) and CRB (dashed lines) are shown for fitted regions of interest ranging from 3 pixels to 41 pixels. Simulations were carried out for two different noise levels, $b = 0$ (pink lines) and $b = 300$ (black lines) and for three different defocus values. The inclination angle was set to $\theta = \pi/6$. A list of all simulation parameters is given in S1 Table in S1 File.

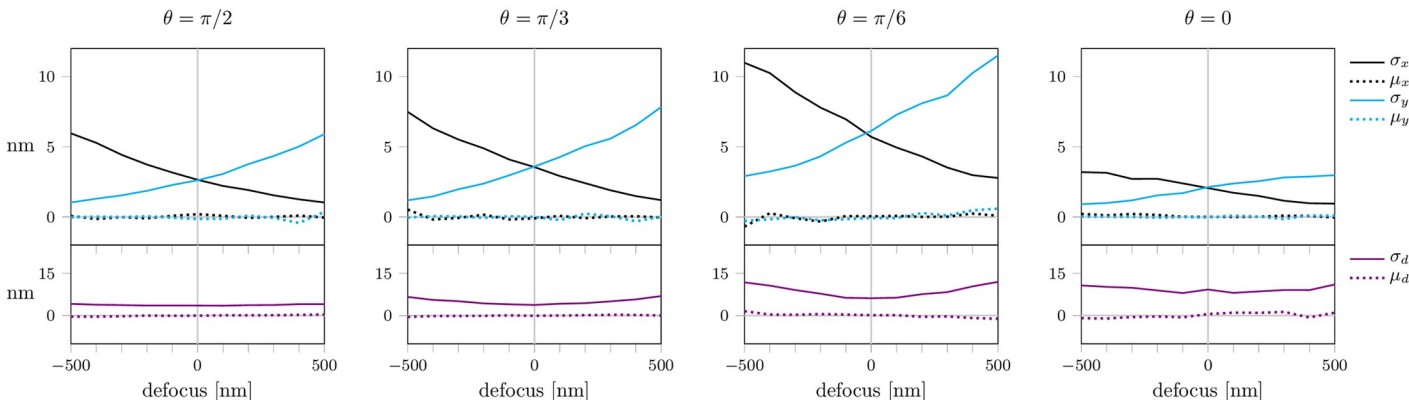

**Fig 7. Influence of uncertainties in dipole orientation.** Simulations and fitting procedure were analogous to Fig 4, except for adding errors in dipole orientation. Errors were distributed normally with standard deviation of 2°. Background noise was set to $b = 100$. A list of all simulation parameters is given in S1 Table in S1 File.

in particular for larger defocusing, leading to a broad PSF in the considered direction. Overall, the localization precision deteriorated, as can be expected with imperfect knowledge of the dipole orientation.

Up to now, we assumed that the photon yield of a fluorophore is independent of the dipole orientation. However, the excitation probability is proportional to $\cos^2(\beta)$, where $\beta$ is the angle between the dipole and the electrical field of the excitation light. Consequently, tilted dipoles will have a reduced photon yield and dipoles almost parallel to the optical axis will have a photon yield close to zero.

In order to examine the influence of reduced dipole excitation on the localization errors, we performed simulations analogous to Fig 4, but with reduced photon yield depending on the dipole orientation (S10 Fig in S1 File). Note that for $\theta = 0$ the excitation probability is zero, hence, we did not include a panel for this dipole orientation in the figure. As in Fig 4 we assumed exact knowledge of the dipole orientation for the fitting procedure. As expected, the results for $\theta = \pi/2$ were identical, whereas the localization precision for tilted dipoles deteriorated. Of note, localization precision still achieved the CRB in all cases.

Finally, in Fig 8 we studied the combined effect of reduced excitation and errors in dipole estimation, which we assumed to be normally distributed as in Fig 7. For high inclination angles, i.e. fluorophores oriented almost parallel to the focal plane, we observed a substantial deterioration of the precision. Interestingly, results for small inclination angles were not strongly affected by such uncertainties, most likely reflecting the insensitivity of the PSF to changes in the azimuthal angle $\phi$.

## Discussion

In this manuscript, we developed a workflow to localize single dye molecules characterized by a fixed transition dipole, as occurring in cryo-SMLM. The problem of biased localization estimates in case of unintended defocus was addressed using deliberate astigmatic distortion of the point spread function, which allows one to achieve a bias-free $(x, y)$ estimate. Additionally, it allows for the determination of the dye's $z$-position with respect to the focal plane. Hence, the method yields a precise determination of single dye positions in $x$, $y$, and $z$. In the literature, two alternative methods were proposed for circumventing orientation-induced $x/y$-bias: in the first approach, a polarization/phase mask in the objective's Fourier plane was shown to abolish the localization bias [33, 35]. Alternatively, also polarization filtering in the emission

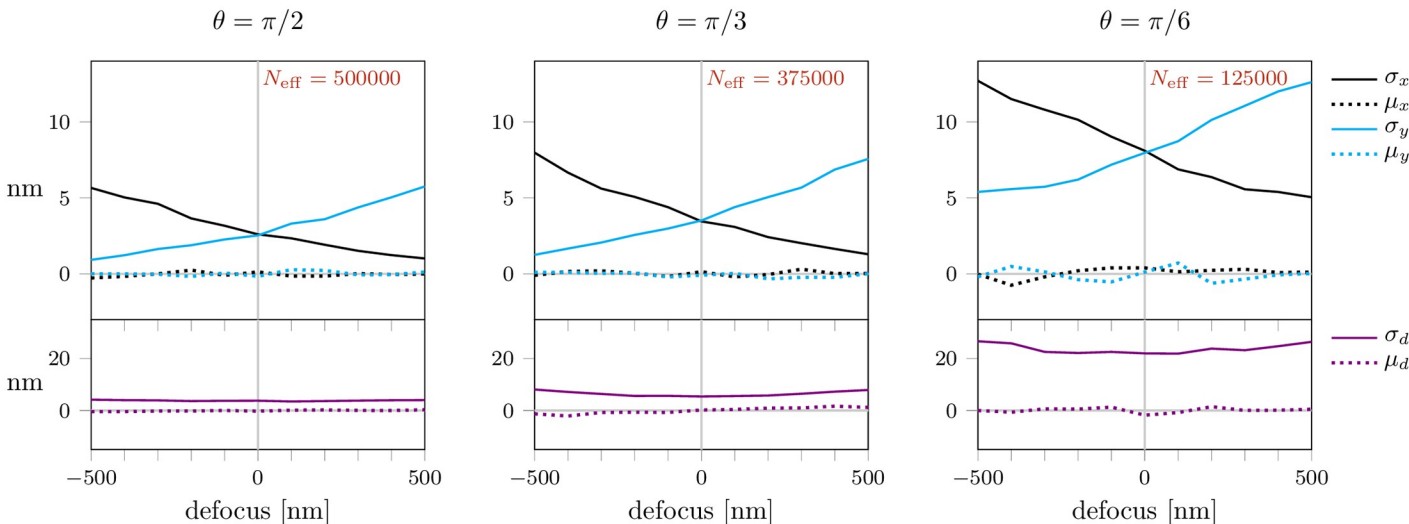

**Fig 8. Combined effect of reduced excitation probability of tilted dipoles, and uncertainties in dipole orientation.** We accounted both for reduced excitation probability of tilted dipoles as well as errors in the determination of dipole orientations; for the latter we assumed a normal distribution with a standard deviation of 2° both for $\theta$ and $\phi$ as in Fig 7. A list of all simulation parameters is given in S1 Table in S1 File.

path was shown to reduce the localization bias [36]. Interestingly, we could show that even the simple approach using astigmatism-based distortion yields unbiased determination of single dye $(x, y)$-positions with an uncertainty limited only by the CRB, with the added value of obtaining the fluorophore's precise $z$-position. Of note, information about the axial position or defocus could also be obtained from other 3D methods, including bi- or multiplane imaging and PSF engineering [37]. A similar approach has been employed by Backlund et al. using a double-helix PSF, where the bias was corrected for using PSF simulations [38]. A recent paper employed a vortex phase mask for simultaneous estimation of the lateral and axial position of the fluorophore as well as the dipole orientation [39].

For the straight-forward application of our method, the following aspects should be considered:

i.  As standard in SMLM, it has to be ensured that the signal is indeed of single dye origin. Many fluorophores tend to show decelerated photobleaching behaviour at low temperatures compared to room temperature [9, 40]. Consequently, it often happens that molecular signals overlap, which would give rise to wrong fitting results. Weisenburger and colleagues suggested a valid strategy by studying the intensity trace of a putative single molecule event [8]: single step transitions between the levels can be identified, and the parts corresponding to single emitter events can be selected for subsequent localization analysis.

ii.  The dipole orientations of the fluorophores need to be fixed and should neither rotate nor wobble. Primarily, our approach is intended for applications in cryo-SMLM, where the mobility of molecules is prohibited. Under room temperature and in aqueous solution, fluorophores typically rotate. In some cases, however, the mobility can be restricted, e.g. by using fluorophores with two attachment sites [41].

iii.  In our manuscript we neglected spatially varying background signal. However, in practice signals from nearby fluorophores or unspecific background may affect the analyzed region. Such scenarios can be approached e.g. by analyzing the evolution of the signals in time. Contributions of signals from nearby fluorophores could be avoided by selecting time

intervals, in which the contaminating fluorophore is in a dark state. In addition, background signals usually fluctuate less in time, and can be subtracted by filters in the time domain [42, 43].

iv. For optimum fit results, the analysis region needs to be large enough to contain the whole signal. As shown in Fig 6, a rather small ROI size of $\sim$15 pixels, corresponding to $\sim$2.5λ, would suffice for optimum localization, yet with the risk of higher fitting instabilities. However, choosing a larger ROI size requires better separation between active emitters.

v. Optimum localization results require *a priori* knowledge on the background noise. This can be obtained from sample regions devoid of any specific fluorescence signal.

vi. Our algorithm assumes that the degree of astigmatism is known. To determine the wavefront aberrations, one could record the three-dimensional PSF of an isotropic single molecule emitter. This could be achieved either by an experiment performed at room temperature on freely rotating dyes, or by summing up the images of multiple fixed emitters with different orientations. Fitting the according Zernike polynomials in Eq (2) allows to extract not only astigmatic distortions, but also potential additional aberrations of the optical setup.

Taken together, we have demonstrated that a rather simple implementation of astigmatic imaging in combination with polarization sensitive excitation allows to avoid localization biases in low NA microscopy, as it is required for SMLM at low temperatures. This may represent another important step towards SMLM applications in structural biology.

## Supporting information

**S1 File. Supplementary table and figures.** The supplementary information contains S1 Table showing the simulation parameters, and S1–S10 Figs.
(PDF)

**S2 File. Software for PSF simulation and fitting.** The latest version of the software is available on GitHub at the following link: https://github.com/schuetzgroup/localizationFixedDipoles.
(ZIP)

## Author Contributions

**Conceptualization:** Fabian Hinterer, Magdalena C. Schneider, Montserrat López-Martinez, Ronny Ramlau, Gerhard J. Schütz.

**Formal analysis:** Fabian Hinterer, Magdalena C. Schneider, Gerhard J. Schütz.

**Funding acquisition:** Alexander Jesacher, Ronny Ramlau, Gerhard J. Schütz.

**Methodology:** Fabian Hinterer, Magdalena C. Schneider, Simon Hubmer, Alexander Jesacher, Ronny Ramlau, Gerhard J. Schütz.

**Software:** Fabian Hinterer, Magdalena C. Schneider, Philipp Zelger, Alexander Jesacher.

**Supervision:** Simon Hubmer, Alexander Jesacher, Ronny Ramlau, Gerhard J. Schütz.

**Writing – original draft:** Fabian Hinterer, Magdalena C. Schneider, Gerhard J. Schütz.

**Writing – review & editing:** Fabian Hinterer, Magdalena C. Schneider, Simon Hubmer, Montserrat López-Martinez, Philipp Zelger, Alexander Jesacher, Ronny Ramlau, Gerhard J. Schütz.

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
