## [Decision Letter · Decision Letter 0]

20 Dec 2021

PONE-D-21-10455Robust and bias-free localization of individual fixed dipole emitters achieving the Cramér Rao boundPLOS ONE

Dear Dr. Schütz,

Thank you for submitting your manuscript to PLOS ONE. After careful consideration, we feel that it has merit but does not fully meet PLOS ONE’s publication criteria as it currently stands. Therefore, we invite you to submit a revised version of the manuscript that addresses the points raised during the review process.

We look forward to receiving your revised manuscript.

Kind regards,

Marek Cebecauer

Academic Editor

PLOS ONE

Journal Requirements:

“The authors were funded by the Austrian Science Fund (FWF): F6805-N36 (RR), F6809-N36 (GJS), P30214-N36 (AJ)”

“The authors were funded by the Austrian Science Fund (FWF): F6805-N36, F6809-N36, P30214-N36”

“The authors were funded by the Austrian Science Fund (FWF): F6805-N36 (RR), F6809-N36 (GJS), P30214-N36 (AJ)”

Reviewers' comments:

Reviewer's Responses to Questions

**Comments to the Author**

1. Is the manuscript technically sound, and do the data support the conclusions?

Reviewer #1: Yes

2. Has the statistical analysis been performed appropriately and rigorously? 

Reviewer #1: Yes

3. Have the authors made all data underlying the findings in their manuscript fully available?

Reviewer #1: Yes

4. Is the manuscript presented in an intelligible fashion and written in standard English?

Reviewer #1: Yes

5. Review Comments to the Author

Reviewer #1: The manuscript "Robust and bias-free localization of individual fixed dipole emitters achieving the Cramer Rao bound" by Hinterer et al. presents an approach that allows unbiased localization of single-emitters at cryogenic temperatures. Generally, fixed dipole moments of single emitters in combination with low NA objectives renders cryo-SMLM imaging challenging. The authors employ classical astigmatism to extract information from defocus and dipole orientation. This approach is shown to allow for an improved fit stability leading to high localization precision.

The authors simulated different data sets and tested different dipole orientations, photon numbers, background signal, fit region size and other key parameters to prove their workflow to be robust for at least high photon numbers.

Overall the paper is well written and the figures are well designed. It is a purely computational work, but with several interesting aspects. The manuscript deserves publication in PlosOne, but I ask the authors to address the following points:

- As the approach of the work treats a particular problem of SMLM experiments at cryogenic temperatures, this should be mentioned in the abstract, probably also in the title.

- Fig. 1: Typo in caption (f)

- Can the authors please justify why they have chosen half a million photons per emission, e.g. by referring to experimental evidence.

- How realistic is the chosen background noise (b = 300 photons standard deviation)?

- Please also simulate photon numbers <10,000 photons, which would be realistic in classical SMLM experiments at RT. Figure S4 could be extended by one or two further panels.

- Based on the proposed method, could any other 3D approaches be advantageously exploited, such as multiplane/biplane imaging?

- Although dipole moments of fluorophores employed in aqueous solution are considered to be freely rotating, are there any situations in which the suggested approach could be of use?

6. PLOS authors have the option to publish the peer review history of their article (what does this mean?). If published, this will include your full peer review and any attached files.

Reviewer #1: No

---

## [Author Response · Author response to Decision Letter 0]

13 Jan 2022

We thank the reviewer for the thoughtful inputs to our manuscript. In the revised version we addressed all raised points. Below, you find a point-by-point reply, including references to the changes in the manuscript. Note that line numbers refer to the manuscript with tracked changes. In addition, we corrected a few further typos in the manuscript.

Reviewer 1

Overall the paper is well written and the figures are well designed. It is a purely computational work, but with several interesting aspects. The manuscript deserves publication in PlosOne, but I ask the authors to address the following points:

Question: As the approach of the work treats a particular problem of SMLM experiments at cryogenic temperatures, this should be mentioned in the abstract, probably also in the title.

Answer: We thank the reviewer for this suggestion. We amended the title and abstract accordingly.

Question: Fig. 1: Typo in caption (f)

Answer: We corrected the wrong panel reference in Fig. 1.

Question: Can the authors please justify why they have chosen half a million photons per emission, e.g. by referring to experimental evidence.

Answer: Indeed, values above 10^6 for the number of obtained photons per fluorophore were reported previously for experiments performed under cryogenic conditions, due to decelerated photophysics. We included a short statement in lines 204–206 of the manuscript and added two references (Li, 2015 and Weisenburger, 2013).

Question: How realistic is the chosen background noise (b = 300 photons standard deviation)?

Answer: For data recorded at the cryo-setup in our own laboratory, we typically observe a background with a standard deviation of 10-12 photons per image. In practice, however, researchers may combine images until photobleaching of the fluorophore. In this case, background noise would increase with the square root of the added images. For example, to obtain a noise level of b = 300 one would need to add 900 frames; hence, the choice of b = 300 represents a rather high estimate for the noise in the data. We included a short statement in lines 206–209 of the manuscript.

Question: Please also simulate photon numbers <10,000 photons, which would be realistic in classical SMLM experiments at RT. Figure S4 could be extended by one or two further panels.

Answer: We included an additional panel row in Figure S4, including four new panels for the different dipole orientations simulated with N = 5000 photons. We refer to the new panel in line 252 of the manuscript.

Question: Based on the proposed method, could any other 3D approaches be advantageously exploited, such as multiplane/biplane imaging?

Answer: We thank the reviewer for this question. Indeed, also other 3D approaches will be helpful for determining the amount of defocus and, hence, also the correct lateral position of the fluorophores. We included a corresponding statement in lines 315–321 of the manuscript

and added references (Deschout 2014, Backlund 2012, Hulleman 2021).

Question: Although dipole moments of fluorophores employed in aqueous solution are considered to be freely rotating, are there any situations in which the suggested approach could be of use?

Answer: Primarily, our approach is intended for applications in cryo-SMLM, where the mobility of molecules is prohibited and, hence, the orientation of fluorophore dipoles is fixed. At room temperature, fluorophore dipoles are typically freely rotating. However, for fluorophores with two attachment sites the mobility could be restricted. Of note, our approach only yields optimal results if the orientation of the fluorophore is fixed and does not allow for wobbling of the dipole orientation. We included a paragraph on the rotation of fluorophores in the Discussion Section in lines 332–337 of the manuscript.

---

## [Decision Letter · Decision Letter 1]

21 Jan 2022

Robust and bias-free localization of individual fixed dipole emitters achieving the Cramér Rao bound for applications in cryo-single molecule localization microscopy

PONE-D-21-10455R1

Dear Dr. Schütz,

We’re pleased to inform you that your manuscript has been judged scientifically suitable for publication and will be formally accepted for publication once it meets all outstanding technical requirements.

Kind regards,

Marek Cebecauer

Academic Editor

PLOS ONE

Additional Editor Comments (optional):

Reviewers' comments:

Reviewer's Responses to Questions

**Comments to the Author**

1. If the authors have adequately addressed your comments raised in a previous round of review and you feel that this manuscript is now acceptable for publication, you may indicate that here to bypass the “Comments to the Author” section, enter your conflict of interest statement in the “Confidential to Editor” section, and submit your "Accept" recommendation.

Reviewer #1: All comments have been addressed

2. Is the manuscript technically sound, and do the data support the conclusions?

Reviewer #1: Yes

3. Has the statistical analysis been performed appropriately and rigorously? 

Reviewer #1: Yes

4. Have the authors made all data underlying the findings in their manuscript fully available?

Reviewer #1: Yes

5. Is the manuscript presented in an intelligible fashion and written in standard English?

Reviewer #1: Yes

6. Review Comments to the Author

Reviewer #1: The authors have answered all of my comments and I can therefore support publication of this manuscript

7. PLOS authors have the option to publish the peer review history of their article (what does this mean?). If published, this will include your full peer review and any attached files.

Reviewer #1: No

---

## [Editor Report · Acceptance letter]

27 Jan 2022

PONE-D-21-10455R1 

Robust and bias-free localization of individual fixed dipole emitters achieving the Cram´er Rao bound for applications in cryo-single molecule localization microscopy 

Dear Dr. Schütz:

I'm pleased to inform you that your manuscript has been deemed suitable for publication in PLOS ONE. Congratulations! Your manuscript is now with our production department. 

Kind regards, 

on behalf of

Mr Marek Cebecauer 

Academic Editor

PLOS ONE